Is painting by elephants in zoos as enriching as we are led to believe?

English Megan 1 2 megan.english@vuw.ac.nz
Kaplan Gisela 1
Rogers Lesley J. 1
1 Centre for Neuroscience and Animal Behaviour, School of Science and Technology, University of New England , Armidale, NSW , Australia
2 Centre for Biodiversity and Restoration Ecology, School of Biological Sciences, Victoria University of Wellington , New Zealand
Vallortigara Giorgio
Electronic publication date: 2014 Jul 1
Publication date: 2014
Volume: 2
Electronic Location ID: e471
Received 2014 May 19; Accepted 2014 Jun 16
Copyright: © 2014 English et al.
Copyright year: 2014
Copyright holder: English et al.
License: This is an open access article distributed under the terms of the Creative Commons Attribution License, which permits unrestricted use, distribution, reproduction and adaptation in any medium and for any purpose provided that it is properly attributed. For attribution, the original author(s), title, publication source (PeerJ) and either DOI or URL of the article must be cited.
License URL: https://creativecommons.org/licenses/by/4.0/

Keywords: Stereotyped behaviour, Asian elephants, Enrichment, Captivity, Painting

Funding: Centre of Neuroscience and Animal Behaviour, School of Science and Technology, University of New England Funding for this study was provided by the Centre of Neuroscience and Animal Behaviour, School of Science and Technology, University of New England. The funders had no role in study design, data collection and analysis, decision to publish, or preparation of the manuscript.

==============================
The relationship between the activity of painting and performance of stereotyped and other stress-related behaviour was investigated in four captive Asian elephants at Melbourne Zoo, Australia. The activity involved the elephant being instructed to paint on a canvas by its keeper in front of an audience. Painting by elephants in zoos is commonly believed to be a form of enrichment, but this assumption had not been based on any systematic research. If an activity is enriching we would expect stress-related behaviour to be reduced but we found no evidence of the elephants anticipating the painting activity and no effect on the performance of stereotyped or other stress-related behaviour either before or after the painting session. This indicates that the activity does not fulfil one of the main aims of enrichment. However, if an elephant was not selected to paint on a given day this was associated with higher levels of non-interactive behaviour, a possible indicator of stress. Behavioural observations associated with ear, eye and trunk positions during the painting session showed that the elephant’s attentiveness to the painting activity or to the keeper giving instruction varied between individuals. Apart from positive reinforcement from the keeper, the results indicated that elephants gain little enrichment from the activity of painting. Hence, the benefits of this activity appear to be limited to the aesthetic appeal of these paintings to the people viewing them.

Introduction

Since the 1960s elephants have been considered intelligent (Rensch, 1957; Gordon, 1966), with significant research about their cognitive abilities taking place in recent years (Shoshani, Kupsky & Marchant, 2006; Bradshaw & Schore, 2007; Hart, Hart & Pinter-Wollman, 2008; Byrne, Bates & Moss, 2009). It is now clear that elephants can solve problems (Foerder et al., 2011), use tools (Chevalier-Skolnikoff & Liska, 1993; Hart & Hart, 1994; Hart et al., 2001), have empathy (Bates et al., 2008; Byrne et al., 2008), recognise human faces (McComb et al., 2014), show the complex behaviour of self-recognition (Plotnik, de Waal & Reiss, 2006; Plotnik et al., 2010; Plotnik et al., 2011) and have a high level of social complexity (Poole, 1996; Schulte, 2000; Payne, 2003; Wittemyer, Douglas-Hamilton & Getz, 2005). It is therefore a reasonable assumption that elephants may be able to learn to paint and they certainly would have no difficulty in using the trunk to direct a paintbrush.

Evidence of their complex communication (Poole et al., 1988; Langbauer, 2000) in stable social relationships (Moss & Poole, 1983; Nair et al., 2009) and their phenomenal memories (Moss, 1988), particularly of spaces and resources, has been demonstrated repeatedly. Questions have been raised how such animals, that are self-aware and remember their own past, are affected by captivity and by traumatic events in the wild (Bradshaw et al., 2005; Jachowski, Slotow & Millspaugh, 2013). Indeed, animals now known to be as complex in their comprehension of their surroundings raise additional questions on how to avoid boredom and stress by confinement. While such establishments as zoos and sanctuaries are expected to provide each species with species-specific enrichment adequate to enhance psychological and physiological wellbeing (Hediger, 1950; Carlstead & Shepherdson, 2000; Mellen & MacPhee, 2005; Veasey, 2006), it is clear that elephants present very challenging problems. They are large in size, and space in zoos may be too limited to be adequate. Their natural feeding habits usually occupy much of the day (and in large quantities) and their browsing habits in the wild usually make them cover large distances (Samansiri & Weerakoon, 2007). Finding compensations for such habits and skills pose almost insurmountable problems and it’s clear to many that such enrichment ought to be more varied and complex than just hiding food (Lair, 1997; Wiedenmayer, 1998). Occupying them in some games or varied routines is certainly a strategy that many zoos have adopted. One of the activities assumed to provide enrichment to captive elephants is allowing them to paint on canvas using coloured paints and brushes under the guidance of zookeepers (Gilbert, 1990; Tennesen, 1998; Rogers & Kaplan, 2007).

According to Forthman & Ogden (1992), a reduction in stress can be realized by creating interesting environments and activities that encourage natural behaviour; for example, by providing animals with opportunities to solve problems, to make choices and to obtain a sense of control over their environment. Enrichment strategies vary widely according to species, staff availability and the ingenuity of their keepers. With a few exceptions (and elephant management is one of them; Sevenich, Upchurch & Mellen, 1998; Shepherdson, 1999; Stoinski, Daniel & Maple, 2000) it has rarely been tested in a rigorous fashion as to whether these measures purported to be beneficial actually are. Many enrichment programs, sometimes of necessity, are highly contrived and find no equivalent in a species’ natural environment. Whether contrived or not, in such cases one might argue that the end justifies the means if it can be shown that there is a measurable benefit to reduce stress and abnormal behaviour, often a consequence of prolonged stress. The question is whether the activity of painting is an activity that might be enjoyable for elephants? While highly imaginative as an idea, of course, as far as we know anything equating painting does not occur in the wild, although the trunk has been shown to be used for many purposes (Shoshani, 1997).

In order to establish whether painting benefits elephants, and is therefore a form of enrichment, some key questions needed to be addressed. Does painting change stress behaviour in any way? If so, is there a difference in stereotyped movements and reduced social interaction on the days when an elephant paints compared to days when it does not? Painting might appeal to the higher cognitive abilities of elephants, as it involves the learnt use of a tool. The tool-using aspect of painting could, therefore, be stimulating to a species that uses tools in the natural environment (Chevalier-Skolnikoff & Liska, 1993; Hart & Hart, 1994; Shoshani, 1997). On the other hand, because the act of painting is not a typical behaviour for elephants it could itself cause stress. A way of assessing stress behaviourally would be to measure stereotyped and abnormal behaviour before, during and after painting, provided any stereotypical or abnormal behaviour had been noted in the elephants to be observed.

Captive animals often develop stereotyped behaviours that are rarely observed in wild or free-ranging animals (Boorer, 1972). A stereotypy has been defined as a repetitive, invariant behaviour pattern with no obvious goal or function (Odberg, 1987; Mason, 1991). Mason & Latham (2004) found that situations in which stereotyped behaviour occurs are usually a sign of poor welfare conditions. However, those individuals performing the stereotyped behaviour, which may be related to an earlier experience of a stressful situation (Schmid et al., 2001), often have lower levels of stress hormones (e.g. cortisol) than those not performing stereotyped behaviour (Mason & Latham, 2004). There is ample evidence that anticipation of being fed (Friend, 1999; Rees, 2004) or of other events important to a captive elephant (Friend, 1999; Kurt & Garai, 2002) may induce stereotypes which occur because arousal levels increase and the animal is unable to do anything but wait until the anticipated activity takes place (Elzanowski & Sergiel, 2006). It has also been found that low temperatures can trigger an increase in stereotyped behaviour if the elephants in question have had a traumatic past (Rees, 2004) suggesting that seemingly unrelated events can alter the expression of distress quite rapidly in elephants.

Elephants display a number of typical stereotyped patterns of behaviour. Weaving behaviour performed by both Asian and African elephants in captivity consists of swaying the head from side to side while transferring the weight from one foreleg to another, and swinging the trunk at the same time. Head bobbing consists of repeated up and down or forward and backward rocking of the head while standing still, and pacing in their enclosures (Kiley-Worthington, 1990; Langbauer, 2000; Rees, 2004). We were interested in measuring whether the prevalence and intensity of these stereotypes and other stress related behaviour varied before and after painting sessions.

Natural body postures in elephants have also been described in great detail and have been known for a long time, indicating that the position of ears, movements and activities of the trunk and the tail may have very specific meanings (Kuhme, 1963; McKay, 1973) and are useful devices for scoring details of an elephant’s state of mind and mood.

Methods

Focal species and enclosures

This study was conducted between October 2007 and February 2008, observing four female elephants housed at Melbourne Zoo. Ethics approval was given by the Animal Ethics Committee at the University of New England (AEC07/096). All four elephants were of Asian origin (Elephas maximus); three from Thailand (Elephas maximus indicus) (hereafter referred to as elephants A, B and C) and one Malaysian (Elephas maximus sumatranus) (hereafter referred to as elephant D). Each elephant had different levels of experience in painting: A (7 years old), B (10 years old) and C (15 years old) had been painting for 2–3 years and D (33 years old) had been painting for approximately 8 years. At the onset of this study, A, B and C had been at Melbourne Zoo for one year and D for 30 years.

The elephants in this study did not constitute a selected sample but represented the entire cohort of participants in painting sessions at Melbourne Zoo at the time; i.e., every elephant was scored that was involved in painting during the study period. The only male housed at Melbourne was at that time not trained in performing painting and was therefore not involved in this study. Which elephants were to perform painting on a given day was decided by the keeper minutes before the elephants were to be moved to the painting enclosure and the choice was based on general demeanour.

The building in which painting took place, also used for medical checks, treatment administration, washing, feeding, and obedience task training, contained four stalls: the stalls in which painting took place were 6 m × 6 m. The elephants were always taken into the building for the painting session in the same pairs (either A/B or C/D), and then separated into different stalls for the painting session. One elephant was given food while the other painted. Each painting session lasted less than 5 min.

Behavioural data collection

Daily behavioural scoring took place at three intervals: from 10.00 am to 11.00 am (3–4 h before the painting session), from 1.00 pm to 2.00 pm (1 h before the painting session) and from 4.00 pm to 5.00 pm (1 h after the painting session). This occurred twice per week. A focal subject was selected for scoring, in pseudo-random order, and its behaviour was recorded in all sessions that day regardless of whether it was selected to paint or not. For each elephant, four days were scored in each of the following conditions: (1) days when the focal elephant painted, (2) days when the focal elephant did not paint and other elephants did and (3) days when no painting took place.

The behaviour was scored using two techniques: (1) videorecorded, and later replayed to score behaviour, and (2) direct observation within 5–20 m of the focal elephant. Minute-interval sampling was used to record various behaviours that occurred for longer durations (for example social interaction, stereotyped or non-interactive behaviour), and event recording was used for behaviour that occurred infrequently and less often (for example, vocalisations).

The following behaviour was scored (see Kuhme, 1963; McKay, 1973; Poole & Granli, 2009; Poole & Granli, 2011):

(1) Interactive behaviour—involving social interactions with conspecifics including visual and tactile displays during affiliation or playful interactions. For example, using the trunk to smell and touch another elephant, ear-flapping, tail-raising without defecation, tail slapping against body or holding ears forward (45°–90° from body).

(2) Non-interactive behaviour was comprised of standing still with the trunk tip touching the ground, ears in neutral position (<45° from body), tail in a neutral position and separate from other elephants. Head is upright and eyes open, in order to differentiate this behaviour from sleeping.

(3) Independent activity included exploratory behaviour, not involving conspecific interaction, such as using the trunk to smell, touch or manipulate an object, placing the ears forward (45°–90° from body) and raising the tail without defecation.

(4) Stereotyped or abnormal behaviour such as weaving, pacing, head-bobbing, trunk-swinging and other repetitive or abnormal behaviours. These were considered to be stereotyped if they were repeated in quick succession and appeared to serve no function related to their external environment.

(5) Waiting at the door or gate to the indoor enclosure where painting took place. This behaiour reflected anticipation and might indicate if an elephant voluntarily put itself in a position nearer to its keepers and to the area where it would paint.

(6)Vocalisations were scored using event sampling. These included chirps, growls, snorts, trumpets and roars.

Behaviour scored during the painting session

Painting involved following instructions from the keeper and performing the task in front of an audience. A canvas was held in front of the elephant at their eye level and at a distance where the elephant could reach it by full or partial extension of the trunk (Fig. 4). Either the keepers, or selected members of the public, chose the colour of the paint. The brush was then handed to the elephant, which it then held with its trunk. Instruction was given by the keeper to the elephant when to start and stop painting, and when to give the brush to the keeper for colour changes. Positive reinforcement was usually given throughout the activity in the form of food and verbal encouragement from the keeper.

Both elephants in the pair that were taken into the building for painting had their behaviour recorded. Painting sessions took place once a week between 2.30 pm and 3.00 pm. Each session lasted between 2 and 5 min, and was a highly structured event. For example, when the elephant was not holding the paintbrush and awaiting the next instruction it was directed to rest its trunk tip on the ground (elephants C/D) or to raise its trunk and rest the tip on the trunk base (elephants A/B). A total of 16 painting sessions were recorded (four per elephant).

Behaviour scored during painting session

(1) Ear positions were recorded to indicate arousal level (Kuhme, 1963; McKay, 1973). The amount of time that the elephant spent with its ears held in particular positions during the painting session was quantified. Positions included, ears forward (45°–90° from body—indicating a high level of arousal), ears neutral (<45° from body) and ears against the body (commonly a sign of apprehension and submission—Kuhme, 1963). Ear flapping was not recorded due to the difficulty in differentiating whether this was a reaction to the activity or for thermoregulation.

(2) Gaze direction indicated whether the elephant was looking at the painting, the audience or the keeper giving commands. Direction of gaze was determined by recording the eye white position. For example, if the white of the left eye was nasal and/ or the eye white of the right eye was temporal, the elephant was viewing the keeper, who always stood on the elephant’s left side.

Stereotyped and abnormal behaviour was not recorded during the painting session due to the short period of time during which there was very little opportunity for the elephants to behave in any way other than that determined by the keeper’s instructions.

Data analyses

Behavioural scores were analysed using the SPSS 16.00 statistical programme. Initially, for each behaviour, we compared replicates (four per elephant) at the three different times of day (morning, midday and afternoon) using a repeated measures General Linear Model (where replicates were the repeated measures and time of day the factor). Then the condition (P = focal elephant paints, NP = no elephant paints and OP = other elephant, not focal elephant, paints) was tested as the factor and time of day as the repeated measure using the repeated measures GLM test. Behaviour recorded during the painting session was analysed, first using a Kruskal–Wallis non-parametric test for heterogeneity, followed by Mann–Whitney U-tests.

Results

Stereotyped and abnormal behaviour

Analysis of these scores revealed that there were no main effects of replicates in any of the three conditions (P, NP, OP) (F3,9 ranged from 0.529 to 1.341 and P ranged from 0.34 to 0.67). There were no interactions between the time of day and replicates (F6,18 ranged from 0.658 to 2.185 and p ranged from 0.09 to 0.68). There was a significant effect of time of day on the frequency of performance of stereotyped and abnormal behaviour but only on the day when the focal elephant was selected to paint (F2,6 = 6.681, p = 0.028; Fig. 1A).

Figure 1 Occurrence of stereotyped and abnormal behaviour.

The mean number of events per elephant (±sem) per 30 min of stereotyped plus abnormal behaviour at (A) the three different times of the day and in the three conditions. P, the elephant scored engages in the activity of painting; NP, days recorded on which no elephant paints; OP, other elephants paint but not the focal elephant being scored. Each elephant was recorded on twelve days (four repeats for each type of day). Note the lower score in the morning of the day when the elephant is selected to paint and the increase at midday and (B) Mean occurrence (±sem) for the individual elephants of stereotyped and abnormal behaviour on days when the focal elephant was not selected to paint. The data are for mean overall number of events per elephant per 30 min (three times a day—morning, midday and afternoon) repeated over four days. A, B, C and D refers to the four elephants studied.

Since ‘replicate’ was found to have no significant main effect or interaction with ‘time of day’, means of the replicates were calculated and used in a further analysis using condition as the repeated measure and time of day as the factor. This analysis revealed that there was a non-significant interaction between condition and time of day (F4,60 = 2.452, p = 0.056). However, the trend towards significance is noted: on the days when the elephants were selected to paint they tended to perform less stereotyped and abnormal behaviour in the morning than on the other days (F2,6 = 6.861, p = 0.028).

Figure 1B shows the variation between individual elephants for the mean occurrence of stereotyped/abnormal behaviour at three different times of the day. The incidence of stereotyped and abnormal behaviour noted in elephants A and B differed to some extent from that of elephants C and D. Stereotypy and abnormal behaviour were found for days when the elephant was not selected to paint as well as on days when the elephant painted.

A repeated measures GLM test comparing the occurrence of stereotyped/abnormal behaviour at different times (morning, midday and afternoon) between elephants for each condition revealed that there was a significant difference between elephants on the days when they were not selected to paint (F6,18 = 7.448, p = 0.008). Bonferroni pair-wise comparisons revealed that B and C were significantly different (p = 0.039) whereas A and D tended to be different but it was not significant (p = 0.059).

Other behaviour

Scores of vocalisations, interactive, non-interactive, independent activity and time spent waiting were analysed separately for the three conditions. As there were no significant effects of the replicates for any of the types of behaviour and no significant interactions between time of day and replicate, the means of the replicates were calculated and used in further analyses using condition as the repeated measure and time of day as the factor. The results are presented in Table 1.

Table 1 Table of results for GLM analysis.

The results of analysis using repeated measures GLM for each behaviour before and after painting sessions. Significant effects are highlighted in bold.

Behaviour	Main effect condition	Main effect time	Interaction b/w
condition & time	
Stereotyped/abnormal	F2,30 = 1.003, P = 0.379	F2,30 = 2.434, P = 0.105	F4,60 = 2.452, P = 0.056	
Vocalisations	F2,22 = 0.007, P = 0.993	F2,22 = 1.674, P = 0.210	F4,44 = 0.062, P = 0.993	
Interactive	F2,30 = 1.003, P = 0.379	F2,30 = 0.885, P = 0.423	F4,60 = 0.994, P = 0.418	
Non-interactive	F2,30 = 1.704, P = 0.199	F2,30 = 2.261, P = 0.122	F4,60 = 2.520, P = 0.050	
Exploration	F2,30 = .422, P = 0.660	F2,30 = 0.466, P = 0.632	F4,60 = 1.196, P = 0.322	
Waiting	F2,30 = 2.091, P = .141	F2,30 = 2.634, P = 0.088	F4,60 = 2.665, P = 0.041	

The only significant results of these analyses were interactions between condition and time for non-interactive behaviour (F4,60 = 2.520, p = 0.050) and time spent waiting (F4,60 = 2.665, p = 0.041). Therefore only these two data sets were examined further.

Non-interactive behaviour

A further repeated measures GLM test to analyse the data for non-interactive behaviour revealed a significant interaction between condition and time (F2,6 = 6.065, p = 0.036). The level of non-interactive behaviour was higher in the afternoon on days when no elephant painted than on days when the focal elephant was selected to paint (p = 0.041; Fig. 2A). The data for non-interactive behaviour were examined further by comparing individual differences between elephants across the three conditions (Fig. 2B). Compared to the other elephants, elephant D displayed much higher levels of non-interactive behaviour on days when no painting took place, especially in the afternoon. This increase from midday to afternoon when no painting took place was shown in all four replications for D.

Figure 2 Non-interactive behaviour.

Mean scores (±sem) of non-interactive behaviour at (A) different times of the day over the conditions are presented. P, NP and OP refer to the three conditions as in Fig. 1 and (B) behaviour for each individual elephant on days when no painting took place. A, B, C and D refer to the four elephants studied.

Overall, the results for the periods outside of the painting session show that elephants C and D performed more stereotyped/abnormal behaviour than elephants A and B, particularly on days when they did not paint. D showed more non-interactive behaviour when no painting took place and more waiting at the door to the indoor area where painting occurs when she was not selected to paint than did the other elephants.

Behaviour during the painting session

During the painting sessions the elephants were given a mean (±sem) of 38 ± 3.49 commands by the keeper. The mean duration of the painting session was 211 ± 41 s.

Direction of gaze during painting session

Elephants A and B, and to a lesser extent C, spent the majority of the session looking at the keeper standing to their left side (Fig. 3A). D looked at the keeper the least often and spent the highest percentage of time looking ahead at the canvas, or the person holding the canvas. A Kruskal–Wallis test revealed a significant difference between elephants for time spent with gaze directed at the keeper (α2 = 9.516, p = 0.023) and gaze directed ahead (α2 = 10.304, p = 0.016).

Figure 3 Gaze direction and ear position during painting.

Mean percentage of time (±sem) spent with (A) gaze following a particular direction. Percentages were calculated by dividing mean time gazing in each direction by total time for which the direction of gaze could be scored. Note the percentage of time that elephant D spent looking ahead. Within each category of looking columns marked ‘a’ differ significantly from those marked ‘b’ (p = < .05). The three sections of columns are, respectively, from the left to the right; looking at the keeper, looking forward and looking at the scorer and (B) Position of the ears during a painting session, presented as percentage of time spent with ears in each position. Significant differences between elephants are shown. a and b indicate significant differences between elephants. a∗ and b∗ indicate that elephant B differed from elephant D in time spent with ears not forward. The four sections of columns, from left to right respectively are; forward ear position (45°–90° from body), not forward (<45° from body), flapping, and back (<10 cm from body).

Figure 4 Image of an elephant painting at Melbourne Zoo.

Position of ears, head and trunk during painting session

Figure 3B shows the percentage of time, during the painting session, for which the ears were held in the different positions. Kruskal–Wallis tests revealed that the elephants differed in time spent with their ears forward (α2 = 8.824, n = 4, p = 0.032) and the ears not forward (α2 = 11.138, n = 4, p = 0.011). Elephant A differed significantly from elephants B (ears back, U = 2.000, p = 0.046), C (ears forward, U = 1.000, p = 0.043, ears not forward, U = .000, p = 0.021) and D (ears forward, U = .000, p = 0.021, ears not forward, U = 0.000, p = 0.018, and ears back, U = 2.000, p = 0.046). B also differed from D (ears forward, U = 1.000, p = 0.043, ears not forward, U = 0.000, p = 0.018).

Discussion

Stereotyped and abnormal behaviour

The scores of stereotyped behaviour tended to be lower in the morning on the day when the elephant painted than on days when no elephants painted. Post hoc enquiries revealed that keepers selected those elephants showing less stereotyped and abnormal behaviour in the morning to paint that day. Hence, our results reflect this practice.

By midday (1 h before painting) the incidents of stereotyped and abnormal behaviour was similar to that performed at midday on the days when no painting took place. Therefore, the increase in stereotyped and abnormal behaviour between morning and midday on the day when the elephant painted may not have been associated with anticipation of the painting session but, instead, indicated a return to levels typical at that time of day. Hence, despite anticipation of certain events (such as feeding and performing), identified as being a major facilitator of stereotyped behaviour in some circus elephants (Friend & Bushong, 1996), there was no evidence that the activity of painting had such an effect on the elephants in our study. Moreover, painting had no significant effect on the incidents of stereotyped and abnormal behaviour performed in the afternoon.

Two elephants performed stereotyped plus abnormal behaviour at similar levels and higher than those of the other two elephants, especially on days when no elephants painted and at all times of the day sampled. This could, in part, be related to differences in past experience, since performance of stereotypies is known to vary according to age, time spent in captivity and handling by owners/keepers (Mason & Latham, 2004).

The types of stereotyped and abnormal behaviour that each elephant displayed varied between individuals; for example, trunk swinging and head-bobbing in C, excessive nipple-rubbing in A, and occasional pacing, trunk swinging and weaving in B. Elephant D, in captivity for the longest period of time of the four elephants tested (<30 yrs), performed weaving more than the other elephants. The weaving behaviour could have developed as a coping response to stressors resulting from a prolonged amount of time in captivity (Mason, 1991). Other causes of stress-related behaviour include removal from a familiar environment, long-distance transport and integration into a group of foreign animals (Schmid et al., 2001). These are all events that the elephants at Melbourne Zoo would have experienced at some time.

It has also been suggested that elephants may suffer from post-traumatic stress syndrome (Bradshaw et al., 2005) in line with the complex cognitive abilities that have been identified in Asian and African elephants. According to Bradshaw et al. (2005), premature or forced maternal separation, insufficient socialisation, and trauma caused by shock can affect psychological, neurobiological, and behavioural wellbeing of elephants throughout early life and into adulthood. These stressful experiences, many of which are likely to be encountered by elephants in captivity, could influence the development and frequency of stereotyped and abnormal behaviour.

Non-interactive behaviour

The analyses of non-interactive data revealed a significant effect of time of day. Higher scores of non-interactive behaviour were recorded in the afternoon on the day when no painting took place than on the other days. This indicates that not being given the opportunity to paint may increase non-interactive behaviour. Since Carlstead (1996) has found that reduced interactive behaviour is a response to stressful situations, our results suggest that not being selected to paint may be stressful. Closer examination of the performance of non-interactive behaviour by each elephant revealed that one elephant (D) showed more non-interactive behaviour than the others, especially in the afternoon when no painting took place. More waiting at the door to the area for painting would have been expected at midday before the painting session if the elephants were anticipating the activity of painting. Since this did not happen, it seems that the elephants were not anxious to take part in the painting session, apart from elephant D, who did wait at the door more than usual and performed stereotyped behaviour when she had not been selected to paint and this behaviour increased after the other elephants that did paint returned to the yard. It may have been a coping response that D had adopted in order to deal with not being selected to paint.

Behaviour during the painting session

During the painting session two elephants (A and B) spent the majority of time looking at the keeper, whereas the others (C and D) spent the majority of time looking forward toward the canvas with their ears facing forward, indicative of high arousal levels (McKay, 1973). It seems, therefore, that elephants C and D were attending to the painting itself to a greater extent than elephants A and B, the latter relying on visual and auditory cues from the keeper. Elephants A and B spent much of the painting session following the command for ‘trunk up’. This position was held while waiting for the next instruction from the keeper and their gaze was usually directed at the keeper at this time. As a reference guide, a chart from Kuhme’s (1963) paper on various head, ear and trunk postures associated with aggression, fear and inhibition in African elephants was used to interpret the postures that the elephants were instructed to hold during the painting session. From Kuhme’s diagram, the extension of the trunk forward to paint with the ears forward resembles the posture held during increased hostility. The posture with the trunk up resembles conflict between fear, arousal and inhibition. Although these postures were adopted on the keeper’s command, they could have influenced the animal’s emotional state. Studies have found that when adopting an emotion-specific posture, humans report experiencing the associated emotions of that posture and their preferences and attitudes are influenced (Niendenthal, 2007). The postures that the elephants are instructed to hold could therefore be influencing how they respond to the painting activity, and to the keepers, which might contribute to the differences in behaviour between the four elephants.

Other variables

There was little evidence possible to collect showing that the keeper had a specific role in the behaviour of the elephants and the researchers had no influence on the zoo schedules. The keepers acting as guides for the elephant’s painting activities were changed each session, and whoever was on duty administered the sequence of tasks in the same manner. Since we know that elephants recognise individual humans and may develop likes and dislikes to specific people, the human influence on such projects could also form an integral part of any study to eliminate the possibility that elephants respond more to the interaction than to the task.

Importantly, it is also questionable what elephants saw while they were painting. A very detailed examination of the structure and pigments of elephant eyes suggests that elephants may be ‘colour blind’. The detailed study by Yokoyama et al. (2005) found that elephants and colour-blind humans (deuteranopes) have identical sets of visual pigments. Potential colour blindness, and the fact that the keepers selected the paint colour, rather than the elephants doing so themselves, may also affect how stimulating the activity is to the elephant. However, perception of the world by elephants is not entirely clear. A more recent study suggests that elephants can classify humans by odour and garment colour (Bates et al., 2007). This involved testing elephants first using two ethnic groups wearing identical red garments, followed by dressing one ethnic group in white and the other in red. The colour red was selected for the study because this is the colour that Maasai men wear, but it is unfortunate in terms of the findings of Yokoyama et al. (2005) because their findings suggest that red is a colour elephants cannot perceive as a separate colour. However, the white would stand out so clearly that it is not a matter of colour but of light intensity, which made it possible for the elephants to clearly distinguish between the two types of garments. While the title of the paper by Bates et al. (2007) suggests that elephants can distinguish people by colour, this is a little misleading and provides little help for the present study.

It is of course also possible to relate the activity of painting in elephants to questions about an aesthetic or general artistic sense in (some) animals. The case for an artistic sense has been pleaded for some songbirds and spontaneous responses to rhythm, thought to be unique to humans, have now been shown in some animals, including elephants (Patel et al., 2009; Schachner et al., 2009; Kaplan, 2009) but far more work, and of a different kind than the context of painting elephants for enrichment, would need to be done to expose the extent and the limits of an artistic sense.

Conclusions

The results of this study showed that all four elephants at Melbourne Zoo perform forms of stereotyped behaviour. Since stereotyped behaviour develops as a response to a stressful environment (Selye, 1973; Mason, 1991; Mason & Latham, 2004), we can propose that the environment in which these elephants live induces stress, or they have been exposed to stressful environments in the past, for example maternal deprivation (Latham & Mason, 2008). The elephants were found to perform different levels of stereotyped/abnormal behaviour and two distinct pairs were apparent: A/B exhibited lower levels of stereotyped behaviour than C/D. Since previous research has found that the performance of stereotyped behaviour is a means of coping in a sub-optimal environment (Mason, 1991; Mason & Latham, 2004), and that individual animals performing more stereotyped behaviour generally have lower levels of physiological stress than those that do not perform this behaviour in the same environment (Moberg, 1985; Koolhaas et al., 1999; Matteri, Carroll & Dyer, 2000), we argue that elephants A and B may be more physiologically stressed than elephants C and D.

One of the key purposes of providing environmental enrichment for captive animals is to reduce the performance of stereotyped behaviour (Swaisgood & Shepherdson, 2005). The level of stereotyped behaviour of the elephants at Melbourne Zoo was largely unaffected by the activity of painting thus not fulfilling a key purpose of enrichment. They all exhibited stress-related behaviours. Elephants are wide-ranging animals that require abundant space and social interaction for their physiological and psychological needs, needs that a zoo environment cannot reasonably fulfil. A key purpose of including enrichment activities for captive animals is to at least reduce stress-related behaviour as well as encourage natural behaviour and stimulate the animals. The activity of painting does not appear to address this need adequately. Our results suggest that painting does not improve the welfare of elephants and that its main benefit is the aesthetic appeal of these paintings to the public and their subsequent sale of which a percentage of funds might be donated toward conservation of the species.

Supplemental Information

Supplemental Information 1 Raw data 1

Click here for additional data file.

Supplemental Information 2 Raw data 2

Click here for additional data file.

The research was made possible by the supportive attitude of Melbourne Zoo giving permission for filming and observing the painting sessions and for providing background information on the elephants. The authors gratefully acknowledge their support.

Additional Information and Declarations

Competing Interests

Author Contributions

Animal Ethics

Field Study Permissions

Professor Lesley J. Rogers and Gisela Kaplan are Academic Editor for PeerJ.

Megan English conceived and designed the experiments, performed the experiments, analyzed the data, contributed reagents/materials/analysis tools, wrote the paper, prepared figures and/or tables, reviewed drafts of the paper.

Gisela Kaplan and Lesley J. Rogers conceived and designed the experiments, analyzed the data, contributed reagents/materials/analysis tools, wrote the paper, reviewed drafts of the paper.

The following information was supplied relating to ethical approvals (i.e., approving body and any reference numbers):

The Animal Ethics Committee at the University of New England approved the study as the study design was purely observation based and required no physical interference with the focal animal: Approval number AEC07/096.

The following information was supplied relating to field study approvals (i.e., approving body and any reference numbers):

Zoos Victoria allowed an admission pass to conduct the study at Melbourne Zoo under Megan English’s student number 220002456. This pass was valid from 30th August 2007 to 30th April 2008 and given by Jan Steele.

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
