# Peer review of "Is painting by elephants in zoos as enriching as we are led to believe?"

_PeerJ, doi:10.7717/peerj.471_

## Round 0.1 · original submission · Minor Revisions

Please provide some comments or rebuttal on comments of Reviewer 2 concerning control group.

Reviewer 1 ·

Basic reporting

The extensive analysis of elephants’ abilities and the comprehensive background that authors discuss make the paper an interesting and enjoyable reading. The contents are clearly described, adequately referenced and I also appreciate the topic of investigation that is quite original. The results show that the activity of painting provides elephants with only little enrichment during their life in captivity, shedding some light onto the need to further investigate the conditions that may improve animals’ welfare in zoos and sanctuaries.

Experimental design

The observations are well conducted with appropriate dependent measures scored and analyzed. I only guess whether a further control could have been relevant, i.e., the recording of the painting activity in absence of the audience. Despite the elephants are described to look at the trainer or to stare at the canvas preferentially, I wonder whether the audience-less environment, controlling for sounds, smells and vibrations from occasional visitors, could have been a crucial condition to observe the effects of the activity per se since their social interactions are not limited to vision. This should be probably discussed in the revised version of the paper. Indeed, elephants may be “happy” of a brief painting practice gaining some enrichment. Being alone in doing so, they can anyway create art for fundraising and still can be visible to visitors by a camera recording and/or transmitting online the elephants’ activity in a separate environment.

Validity of the findings

Although the authors are aware of the limitations of their study, which concerns 4 animals only, very different one from the other in amount of time spent in captivity and in time spent painting I wonder whether simple physiological measures of stress (i.e., heart rate for instance) could have been easily obtained, increasing the strength of the main findings.
Also, I guess it would be worth reporting that another artistic activity (i.e., listening to music or playing an instrument) seems to be a more appropriate activity to enrich elephants' lives. The suggestion arises from evidence that elephants entertain spontaneously to music (Schachner et al., 2009) and if in the condition of playing instruments of appropriate dimensions they spontaneously initiate the activity of playing (as reported by the authors of the musical pieces composed for the Thai Elephant Orchestra).

Additional comments

I highly recommend to accept this paper not only because well conducted despite obvious difficulties, but also because it is important to put some limits through rigorous investigation to the activities that may represent a real condition of enrichment and activities that are instead purely human and produce humans' amusement only, but increase stress-like behaviors in animals.

Reviewer 2 ·

Basic reporting

No comments

Experimental design

It is unclear how the elephants were selected for the painting sessions.
It is unclear how and when elephants could understand they were selected for painting.
Authors observed the behaviour of four female elephants — why have they excluded males from their study? Would they expect a similar or a different behaviour?
My major point concerns the lack of a control group composed of animals that undergo a control task. I think it could be interesting to compare the measures reported here with those obtained with animals employed in a control task. Further, do the Authors have an idea of the baseline level of stereotyped behaviours in elephants that do not paint? It would also be relevant to compare the measures of these different groups.

Validity of the findings

P=0.056 is insignificant. Therefore, I suggest the Authors leave out this effect — or at least to not draw strong conclusions on the basis of such results.

Additional comments

I have really appreciated the topic discussed in this article, but I strongly recommend a comparison of these results with those of animals that undergo a control task.

---

## Round 0.2 · accepted · Accept

I read your revised ms and response to reviewers and it seems to me that you have done an excellent job in revising the ms and have replied adequately to all comments of the reviewers.